

# 2021 Alaska Earthquake: entropy approach to its precursors and aftershock regimes.

Eugenio Vogel[1,2,3], Denisse Pastén[4], Gonzalo Saravia[5], Michel Aguilera[6], and Antonio Posadas[7,8]

[1]Departamento de Física, Universidad de La Frontera, Casilla 54-D, Temuco, Chile
[2]Center for the Development of Nanoscience and Nanotechnology (CEDENNA), 9170124 Santiago, Chile
[3]Facultad de Ingeniería, Universidad Central de Chile, Santiago 8330601, Chile
[4]Departamento de Física, Facultad de Ciencias, Universidad de Chile, Santiago, Chile
[5]Los Eucaliptus 1189, 4812537 Temuco, Chile
[6]Universidad Tecnica Federico Santa María, Valparaíso, Chile.
[7]Departamento de Química y Física, Universidad de Almeria, 04120 Almeria, Spain.
[8]Instituto Andaluz de Geofísica, Campus Universitario de Cartuja, Universidad de Granada, 18071 Granada, Spain.

**Correspondence:** Denisse Pastén denisse.pasten.g@gmail.com

**Abstract.** For the first time, an entropy analysis has been made in Alaska, a seismic-rich region located in a subduction zone that shows a nontrivial behavior: the subduction arc changes the seismic activity from the eastern zone to the western zone, showing a decrease in this activity along the subduction. We analyze this zone through the Tsallis entropy and the mutability (or dynamic entropy). The data set used for this analysis was measured between 2000 and 2023, considering 19 549 seismic

events. We have found an agreement between the results for the two entropies. We have followed the value of the $q$ parameter of the Tsallis entropy, finding values between 1.70 and 1.85, in concordance with values previously found in other seismic regions of the planet. Also, the values of the $q-$parameter tend to decrease over time but show an increase before the main earthquakes. Mutability also shows a tendency to decrease over time and an increase previous to the main earthquakes. To deeply analyze this zone, we divided the region into four sub-zones. The results show how mutability is able to identify the

seismic activity in each zone. This study shows how an entropy approach can shed light on understanding the seismicity in subduction zones.

## 1 Introduction

The seismic background in Alaska has been a source of questions and studies for the last decades. The subduction of the Pacific Plate under the North American Plate produces the Aleutian trench zone, which runs almost parallel to the arc of islands running

on the Southern part of Alaska pointing to Asia to the west. The Aleutian arc has an extension of approximately 3000 km, from the Gulf of Alaska (east) to the Kamchatka Peninsula (west) (USGS, 2024). Like other subduction zones on the planet, it is a geologically active area in both underground seismicity and surface volcanic eruptions. The Pacific Plate is moving in a northwest direction at average rates of 60 mm/year in the east, and 76 mm/year in the west. The rich activity of this zone lies in the different tectonic sources of its seismicity (Martin-Short et al., 2018). On one hand, the seismicity of the central and the

eastern portions of the arc are both greater than the one of the western portion. In the latter, the seismic activity is shallow, and





the volcanic activity decreases, in comparison with the central and eastern zones of the Aleutians. At least 12 large earthquakes have occurred in the eastern and central areas of the arc in the last century, with magnitudes greater than $M_w 7.5$.

So, the Alaska-Aleutian Region has a rich history of large earthquakes: the Shumagin Island in 1938 ($M_w 8.2$); the Aleutian Islands in 1946 ($M_w 8.6$); the Queen Charlotte Islands in 1949 ($M_w 8.1$); the Lituya Bay in 1958 ($M_w 8.2$); the Prince William Sound in 1964 ($M_w 9.2$) which is the second largest earthquake worldwide since there are reliable registers; the Rat and Near Island segment in 1965 ($M_w 8.7$) (Qu et al., 2022). In recent years, at least four large earthquakes have occurred in this zone: the Simeonof earthquake in July 2020 ($M_w 7.8$); the Alaska Peninsula earthquake in October 2020 ($M_w 7.6$); the South Alaska Peninsula in July 2021 ($M_w 8.2$), almost exactly one year after the previous and the late aftershock of 2023 ($M_w 7.2$). This high seismic activity in the Alaska Peninsula in the last three years deserves dedicated attention, which is the purpose of this article. Researchers have studied the Alaska subduction zone from different points of view for several years (Biswas et al., 1986; Doser and Rodriguez, 2011; Smith and Tape, 2019; Daly et al., 2021; Qu et al., 2022). The present article follows a different line to previous works as we will present an analytical approach based on the on the variations of the entropy in catalogs of magnitudes and intervals.

As has been shown, the analysis of seismicity through information theory has deepened our understanding of this system out of equilibrium. Studies based on entropy have been able to follow the time evolution of seismicity and they are especially useful in subduction tectonic zones (Sigalotti et al., 2023; Skordas et al., 2020; Varotsos et al., 2018; Vallianatos et al., 2015; Telesca, 2010, 2011; Vogel et al., 2017b; Posadas et al., 2022, 2023; Pasten et al., 2023). There are different ways to define and use entropy. In this study, we will use two specific approaches: Tsallis entropy and mutability (a form of dynamical entropy). To study the time evolution of the seismic data, we will use an enumeration of the events; this method can be compared to the well-known concept of natural time (Varotsos et al., 2011, 2019).

The aim of the present analysis is to find patterns in the data sequence that could lead to understanding the following aspects of the process: a) entropic activity of the zone or sub-zone that can be an indicator of seismic risk; b) parameters serving as indicators of seismic risk; c) behavior of the data sequence during the earthquake and immediate aftershocks; d) recovery to the "normal" or previous seismic activity following the aftershock period.

The next section devotes itself to methods, starting with the extraction of data from the USGS catalog and filtering mechanisms. Then, the researchers quickly review Tsallis entropy and mutability, primarily referring to previous publications to avoid repetition. Section 3 is where the authors focus on results and discussions, while they dedicate Section 4 to conclusions. We include an Appendix to show how the dynamical entropic analysis can be performed on the real-time intervals between successive events.



## 2 Methodology

### 2.1 Data

We make use of the catalog of the United States (USGS) on the website https://earthquake.usgs.gov/earthquakes/search/. The process starts by defining the geographical area of interest, which, in our case, is determined by the geographic coordinates: $53.5°$ (N) $<$ Latitude $< 57.5°$ (N) and $-161.0°$. In this way, we include the most important recent earthquakes in Alaska.

Next, we need to specify the period under study. Since we want to study signs of previous activity, we will extend it from January 1, 2000, to December 31, 2023. The hypocenter considers an arbitrary initial depth limit of 100 km, while we collect all seisms with a magnitude equal to or over 1.5 ($M_w \geq 1.5$). This meant that we started this study with 19,549 seisms, and we will conduct further filtering below. Table 1 lists the main four earthquakes for the present analysis, labeled A, B, C, and D from now on.

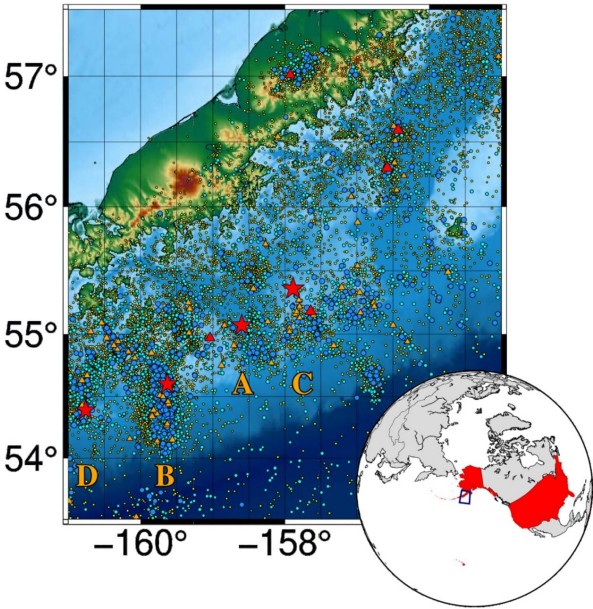

**Figure 1.** Illustration of the seisms with epicenters within the volume between $53.5°$N and $57.5°$N, $155°$W and $161°$W and up to 70 km deep. The largest stars correspond to the 4 earthquakes listed in Table 1 and they are labeled A, B, C, and D, following the order of occurrence. Other symbols correspond to seisms of increasing magnitude according to the following color code: $2.0 \leq M_w \leq 3.0$ yellow; $3.1 \leq M_w \leq 4.0$ cyan; $4.1 \leq M_w \leq 5.0$ blue; $5.1 \leq M_w \leq 6.0$ orange; $6.1 \leq M_w \leq 9.0$ red.

Fig. 1 shows the geographical area defined by this search, representing the magnitude of the seisms with the following color codes: $2.0 \leq M_w \leq 3.0$ yellow; $3.1 \leq M_w \leq 4.0$ cyan; $4.1 \leq M_w \leq 5.0$ blue; $5.1 \leq M_w \leq 6.0$ orange; $6.1 \leq M_w \leq 9.0$ red.





| Date | $M_w$ | Latitude | Longitude | Depth (km) | Label |
|---|---|---|---|---|---|
| 2020.07.22 | 7.8 | 55.07 | -158.60 | 28.0 | A |
| 2020.10.19 | 7.6 | 54.60 | -159.63 | 28.4 | B |
| 2021.07.29 | 8.2 | 55.36 | -157.89 | 35.0 | C |
| 2023.07.16 | 7.2 | 54.39 | -160.76 | 25.0 | D |

**Table 1.** Four main seisms within the zone of interest.

Large red stars denote the four largest seisms tabulated in Table I, identified by the letters A, B, C, and D, following the order of happening.

Fig. 2 reports the Gutenberg Richter analysis for these data. Following the principle of maximum curvature, we can pick a magnitude of $M_w 2.1$ as the minimum magnitude compatible with a distribution characterized by a linear decrease in this diagram: a red vertical line marks this in the plot.

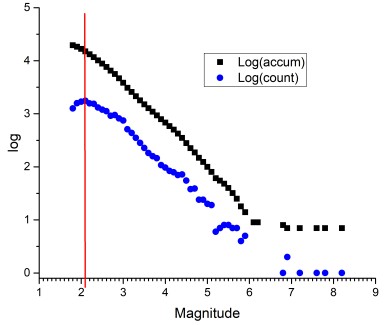

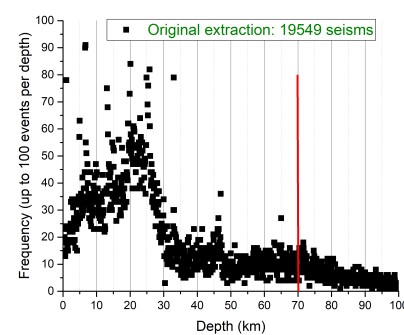

**Figure 2.** Left) Gutenberg Richter plot for the 19 543 seisms initially considered in this paper; the red vertical line sets the minimum magnitude at 2.1 for further analysis. Right) Depth frequency for the 19 543 original seisms; the red vertical line sets the maximum depth at 70 km.

Depth fixes the next filter. The most devastating effects of earthquakes in terms of human life come from the tsunamis produced by large enough earthquakes. Correction 1: This needs to be near the ocean bottom, specifically not a deep hypocenter. As in a previous study (Posadas et al., 2023) we set the maximum depth for the present study at 70 km, which includes over 80% of the originally collected seisms.

After applying these two filters, we have a definite collection of 15 011 seisms (representing 70.96 % of the original sample). We present the time distribution of these seisms in Fig. 4. To the left, we appreciate how the activity of this region continuously increased irregularly since the beginning of the century. To the right, the lack of activity prior to the largest seisms is more clear upon zooming to the last 4 years.





At this point, we can follow two complementary ways to analyze these data through entropy: a) magnitude sequence, and
b) sequence of intervals between consecutive seisms. We have chosen the former since entropy is more easily associated with
energy, even since the early thermodynamics courses. However, we will include some results using the latter in the Appendix.

In Fig. 3, the depiction presents the magnitude of each of the 15 011 seisms at the time of their production. The left panel
renders the full picture, while the right panel focuses on the last 4 years of the interval. At first glance, we notice the irregularity
of the sequence, as we can expect from this kind of stochastic phenomenon. In addition, a clear lack of activity precedes the
earthquakes with the larger magnitudes.

Now, we can treat the sequence of $N = 15\,011$ magnitudes dynamically in overlapping windows of $W$ consecutive events.
Thus, the first window includes seisms from $n = 1$ to $n = W$, the second window from $n = 2$ to $W + 1$, and all the way to
$n = N - W + 1$ to $N$. The data of these mobile windows will be used by the two entropies calculated below: Tsallis entropy
$S_q(t)$, and 'mutability $\zeta(t)$. The time $t$ is given by the time of the last element in the corresponding dynamical window $W(t)$.

## 2.2   Tsallis Entropy

Within each window of $W$ registers, we find a distribution of data where $f_i$ corresponds to the frequency of magnitude $M_i$.
Then the probability $p_i$ of getting the value $M_i$ is simply given by:

$$p_i = \frac{f_i}{W} \,, \tag{1}$$

where the sum extends to all $\Omega$ possible states $i$.

It immediately follows that:

$$\sum_{i=1}^{\Omega} p_i = 1 \,, \tag{2}$$

adding over all the $\Omega$ accessible states within that window.

The average magnitude within the window ending at the $n - th$ register $< M_n >$ can be directly obtained from the registers
in the window, but it can also be expressed as

$$< M_n >= \sum_{M_i > M_0}^{M_{max}} p_i M_i \,, \tag{3}$$

where $M_0$ and $M_{max}$ are obtained from the Gutenberg Richter distribution. Namely, $i_0 = 2.1$ and $i_{max} = 8.2$ for the present
case.

And following Shannon (Shannon, 1948) we can write the entropy $H$ omitting Boltzmann's constant as:

$$H = - \sum_{i > i_0}^{i_{max}} p_i \, log(p_i). \tag{4}$$

Eqs. 2, 3 and 4 are now treated analytically in the following way: states, and their probabilities $p$ are formally treated
as continuous variables, so integral replace sums. Lagrange multipliers combine equations 2, 3 and 4 to get conditions for





maximum entropy. (The algebraic details have been recently given somewhere else (Posadas et al., 2023) so we omit them here.) This leads to a $q$ value determined by the distribution within each window which then leads to the Tsallis entropy by

means of (Sotolongo-Costa and Posadas, 2004; Posadas and Sotolongo-Costa, 2023). In the present paper we make use of the second approach. Once $q$ is determined the Tsallis entropy calculation is straightforward:

$$S_q = \frac{1}{q-1}\left(1 - \sum_{i=1}^{\Omega} p_i^q \,.\right) \tag{5}$$

Boltzmann-Gibb's entropy is recovered in the limit $q \to 1$ but for most seismic zones $q > 1.0$.

### 2.2.1  Mutability

We go back to the original vector file, storing the magnitudes of the sequential 15 011 seisms under scrutiny. To each of the successive time windows of $W$ information recognizer wlzip applies to get mutability $\zeta$ defined as (Vogel et al., 2017a; Pasten et al., 2023):

$$\zeta(n) = \frac{w_n^*}{w_n} \,, \tag{6}$$

where $w_n^*$ is the weight in bytes of the map created by wlzip of the time window at position $n$, whose weight was $w_n$.

Upon creating the map of each window, the algorithm generates a histogram of the frequencies as a byproduct, enabling the calculation of Shannon's entropy. However, the mutability bears dynamic information, which is lost in the simple histogram. So we will continue the analysis with $\zeta(n)$. We recently provided all technical details concerning this process in (Pasten

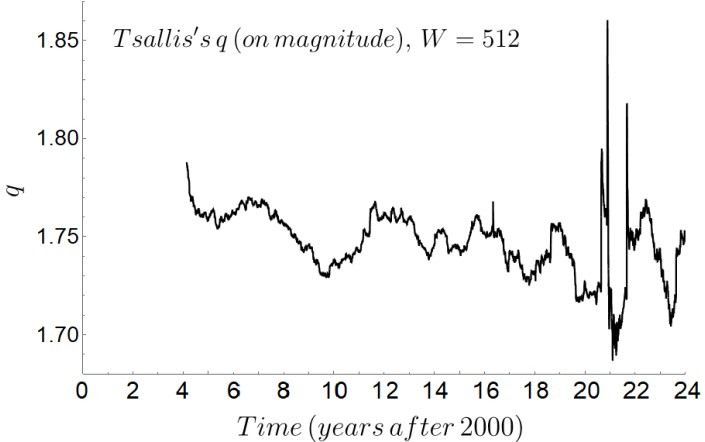

**Figure 3.** Variations of the Tsallis $q$ value along the period of study for overlapping windows of $W =512$ consecutive events.

et al., 2023), so we will not repeat them here. Let us just remind that mutability reaches its minimum value for repetition of information (magnitude of the seisms in our case). Before a major earthquake occurs, the subduction mechanism is nearly

halted, allowing only small advancements. These small advancements result in medium to low magnitudes that are very similar




to each other, enabling better compression and a lower value of mutability. When the rocks finally break down, they produce a large quake with a high magnitude, followed by a dispersion of magnitudes that differ significantly from each other. This sudden change in magnitudes causes the mutability to rise abruptly, resembling upward needles in the diagrams below.

## 3 Results and Discussion

### 3.1 Tsallis entropy

From Eq.5 we realize that Tsallis entropy is determined by the $q$ value obtained from the data distribution. Fig. 3 presents the variations of $q$ with time for the region under study, for $W = 512$ along the 24 years. The texture of the curves varies according to the window size, but the ranges do not vary much. Actually, fluctuation around 1.75 agree well with previously calculated $q$ values for other earthquakes in other regions of the world (Sotolongo-Costa and Posadas, 2004; Silva et al., 2006; Telesca and

Chen, 2010; Telesca, 2011; Darooneh and Mehri, 2010; Valverde-Esparza et al., 2012).

We recently provided all technical details concerning this process in (Pasten et al., 2023), so we will not repeat them here. Let us just remind that mutability reaches its minimum value for repetition of information (magnitude of the seisms in our case). Before a major earthquake occurs, the subduction mechanism is nearly halted, allowing only small advancements. These small advancements result in medium to low magnitudes that are very similar to each other, enabling better compression and a

lower value of mutability. When the rocks finally break down, they produce a large quake with a high magnitude, followed by a dispersion of magnitudes that differ significantly from each other. This sudden change in magnitudes causes the mutability to rise abruptly, resembling upward needles in the diagrams below.

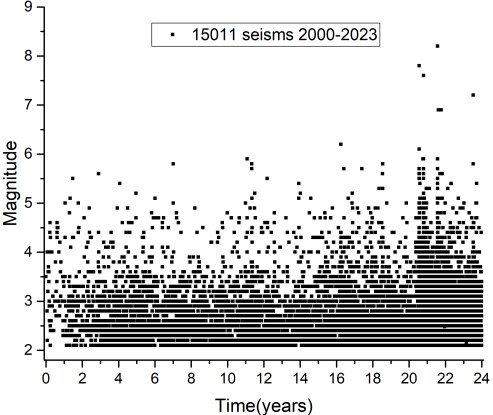
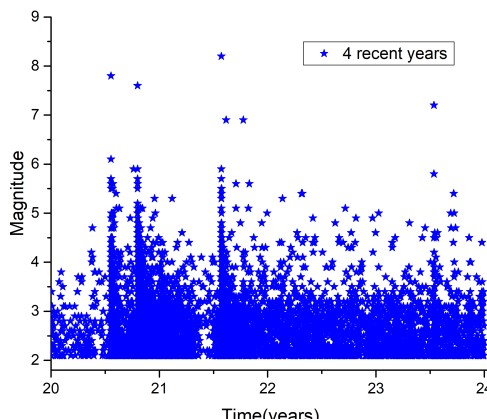

**Figure 4.** Left) Magnitude of the sequence of 15 011 along the years 2000-2023. Right) Detail of the years 2020-2023 where the seisms A, B, C, and D are clearly appreciated by there magnitudes over 7.0. Noteworthy is the decrease in seismic activity just before these large earthquakes.





Sotolongo-Costa introduced a method (which we essentially follow here) and reported $q$ values in the range 1.60 to 1.64 for Southern Spain and 1.65 for California (Sotolongo-Costa and Posadas, 2004). From there this method and its variations have

been applied to different regions of the world. Thus, we find the work of Silva et al. (Silva et al., 2006) reporting $q$ values of 1.60, 1.63, and 1.71 for data from Brazil, the USA, and Turkey, respectively. Then, Dahoornet and Mehri (Darooneh and Mehri, 2010) reported values of 1.78 and 1.81 for seismic areas in Iran and California respectively. Luciano Telesca studied the L'Aquila seismic region by non extensive entropy and reported q values of 1.48 in the early period increasing to 1.74 and 1.70 later on (Telesca, 2010). A similar analysis was then done for the seismicity of Taiwan reporting $q = 1.685$ (Telesca and

Chen, 2010). For the seismicity of California, a variation of these methods yielded $q = 1.54$ (Telesca, 2011). The seismicity of Mexico was also explored by nonextensive methods, and $q$ values of 1.7, 1.69, 1.63, and 1.64 were reported for different areas (Valverde-Esparza et al., 2012).

The differences among the $q$ values in the brief review of the previous paragraph can be due to both variations in the method and different geological characteristics of the different zones picked for this sampling. Nevertheless, two facts are clearly

established 1) $q$ is much larger than 1.0, so non-extensive treatments are necessary and 2) The ranges reported by Fig. 3 are of the same order of magnitude as other active seismic zones.

Fig. 5 reports the Tsallis' entropy calculated for windows $W = 512$ and $W = 1024$, for the 15011 seisms covered by this report. The left-hand side presents the variations along the 24 years of this study, while the right-hand side zooms on the last 4 years, where most of the important activity shows up.

Fig. 5 presents the average value of the Tsallis entropy over windows with W=512 and W=1024. The relevant seisms A, B, C, and D given in Table 1 are also marked by the symbols given in the upper part of the plot. The left panel covers the full 24 years, while the right panel concentrates on the last 4 years. Independently of the time windows, it can be noticed that at the precise moment the large earthquakes are produced, the Tsallis entropy suddenly decreases. The larger window ($W = 1024$) retains most of the characteristics of the function but it has two disadvantages: i) It shows less texture than the

one with $W = 512$, and ii) it begins to show results after 5 years of gathering data to reach the 1024 seisms; such delay is not desirable to detect risk in advance. So we will prefer $W = 512$ when possible, accepting $W = 256$ in cases with poor statistics.

The average magnitude tends to be minimized before a large earthquake. Either 256 or 512 seems to yield similar results, however, larger windows present a noticeable delay which is not desirable if we want to advance towards detecting seismic risk areas which is one of our goals.

Tsallis entropy deals with the probability of populating the states defined by the underground motion of the plates within the volume described by the 4 degrees in latitude, 6 degrees in longitude, and 70 km depth. This we cannot measure, but we can process the distribution of magnitudes product of the interplay of the interactions of the huge stalactites and huge stalagmites collapsing into the gouge area, breaking or lifting rocks, enlarging or closing the "empty" space in between. Actually, it is empty of solid matter but filled with compressible gases which do not contribute significantly to the entropy. It is the sizes and

distributions of plates, stalagmites, stalactites, and the space in between that mainly determines the entropy value. Thus, if there is complete emptiness in this volume, the entropy is zero by definition. The changes in entropy are mainly due to variations in the empty regions as the solid masses collide and slide away. Each window reflects this in its seismic sequence. Then, when


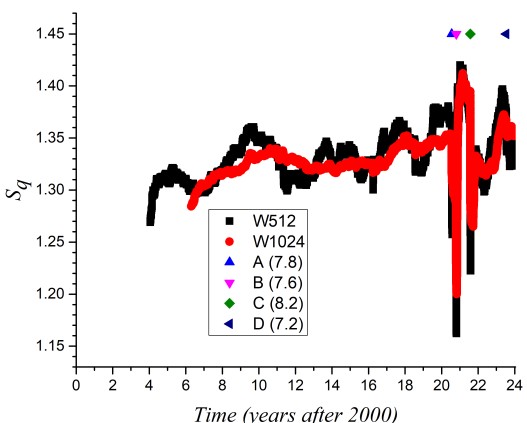 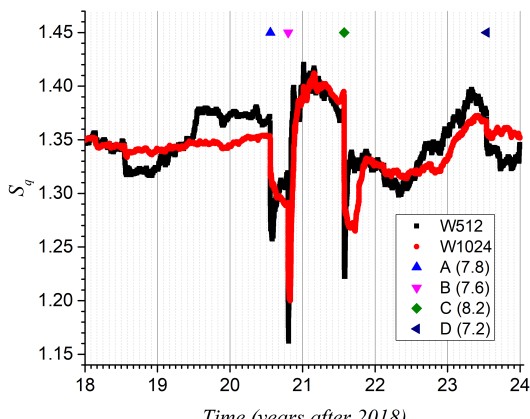

**Figure 5.** Left) Tsallis' entropy for the seismic sequence from 2000 to 2023 using mobile windows of $W = 512$ events (black) and 1024 events (red). On the top line, the largest seisms A, B, C, and D of Table 1 are clearly indicated. Noteworthy is the long run increase in entropy during the years previous to the large earthquakes. Right) Zoom on the recent years of the same data.

the system is stuck, the opposing plates struggle to repel each other, increasing the empty space in between and increasing the Tsallis entropy. When an earthquake is unleashed, the plates come suddenly closer again, closing the empty space and causing an abrupt decrease in Tsallis entropy.

### 3.2 Mutability

Fig. 6 reports the variations of mutability for the same windows of 512 and 1024 consecutive events used with Tsallis entropy plotted in the previous figure. The left-hand side reports the mutability for the entire interval 2000-2023, while the right-hand side zooms on the last 4 years, which is where the large earthquakes concentrate. Mutability decreases previous to a large earthquake because small seisms of restricted magnitude dominate the sequence. The general slope of decreasing the mutability value is clear for the $W = 1024$ window, with the disadvantage of the delay in time between cause and effect. The absolute minimal mutability after the $M_w 7.6$ earthquake, whose aftershock regime was suppressed, is notorious. This is almost an announcement of the 8.2 earthquake, which shows here as a "needle" pointing upwards in the mutability function.

### 3.3 Tsallis entropy and mutability

The agreement between Tsallis entropy and mutability is clearly reflected in Fig. 7. This brief window of $W = 256$ events is noisier than previous ones, which actually underlines the coincidences between these two functions. Despite being calculated by completely different algebraic procedures, the information content embedded in the distribution of magnitude values leads to simultaneous increases of Tsallis entropy and decreases of mutability as the large earthquake approaches. Downwards needles

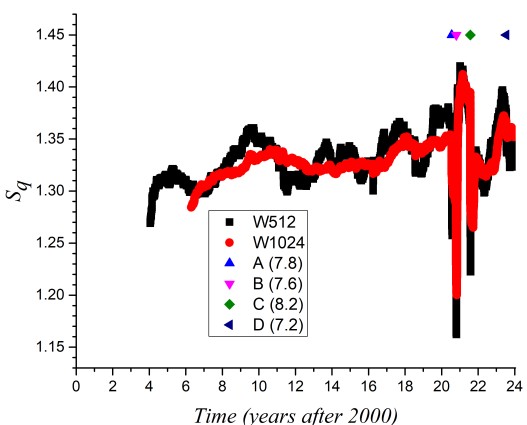
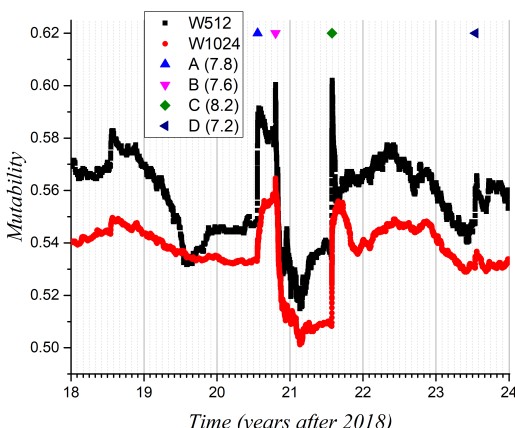

**Figure 6.** Left) Mutability on magnitude for the seismic sequence from 2000 to 2023 using mobile windows of $W = 512$ events (black) and 1024 events (red). On the top line, the largest seisms A, B, C, and D of Table 1 are clearly indicated. Noteworthy is the long run decrease in entropy during the years previous to the large earthquakes. Right) Zoom on the recent years of the same data.

from Tsallis entropy coincide completely with upwards needles from mutability. Noteworthy is the weak response of both
functions to the $M_w 7.2$ earthquake in 2023.

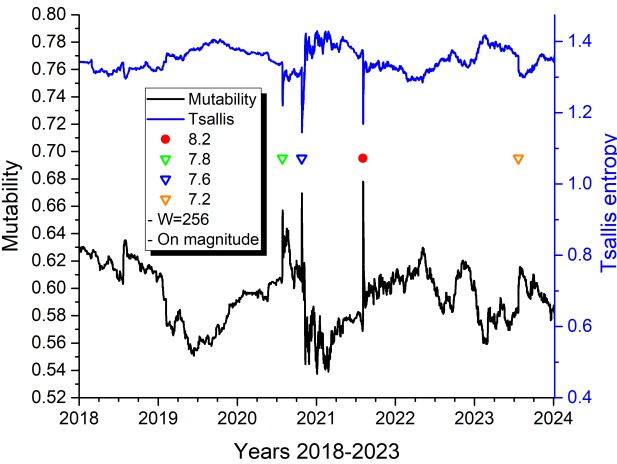

**Figure 7.** Tsallis entropy and mutability on magnitude along the years 2018-2023, for windows of 256 events.

This agreement was even more significant with the seismic activity a few days previous to the $M_w 8.1$ earthquake in front of Iquique (Chile) in the year 2014 (see Fig. 7 of Ref. (Pasten et al., 2023). Both mutability and Tsallis entropy revealed an





incoming seismic activity, increasing from days to minutes before the large quake. In the present case, we did not observe this effect for the recent Alaska earthquakes, and for each of the 4 events reported in Table 1, we found a sudden increase

(decrease) in mutability (Tsallis entropy) with no previous announcement. Obviously, the underground for these two zones can be completely different, and no generalizations are possible. But even for these 4 seisms reported in Table 1 each earthquake can reflect different underground dynamics. This is an excellent opportunity to investigate this point for 4 seisms close in geography and time. A careful look at Fig. 7 reveals that mutability shows more texture than Tsallis entropy for the same data. For this reason, we carry on with calculations of mutability only since these results have all the information that the time series

can yield.

## 3.4 Sub-zones

We begin by defining 4 non-overlapping sub-zones on the map given in Fig. 1. Each subzone experienced an earthquake with a magnitude of over 7.0 near its center and included a large cluster of smaller seismic events around it, from January 1, 2020, to December 31, 2023, and up to a depth of 70 km. Table 2 summarizes the geographic definitions of the 4 sub zones. We choose

to present the time in number of days to better focus on the period of interest (there is a total of $18 \times 365 + 6 \times 366 = 8766$ days in the period under consideration).

| Seism | $M_w$ | Date | Latitude | Longitude | # events |
|-------|-------|------|----------|-----------|----------|
| A | 7.8 | 2020.07.22 | 54.0 to 55.0 | -160.5 to -161.0 | 929 |
| B | 7.6 | 2020.10.19 | 54.5 to 55.5 | -157.3 to -158.2 | 4021 |
| C | 8.2 | 2021.07.29 | 54.0 to 55.0 | -159.5 to -160.0 | 629 |
| D | 7.2 | 2023.07.16 | 50.0 to 50.5 | -158.0 to -158.5 | 908 |

**Table 2.** Definition of the sub-zones studied separately by means of mutability. The first column gives the label of the main seism; the second and third columns give its magnitude and its date respectively. The fourth column gives the range in latitudes while the fifth column gives the range in longitudes. The sixth column gives the number of seisms for each subzone.

The first important difference among the sub-zones is the total number of seisms collected over the 24 years for each one of them. They go from 629 for sub-zone C (containing the largest $M_w 8.2$ earthquake!) to 4 021 for the sub-zone B. Analyses based on $W = 1024$ are not possible while those based on $W = 516$ are nearly meaningless. So we will use $W = 256$ for the

sub-zone analysis.

The time distribution of the earthquakes shows apparent differences in their dynamics. Fig. 8 shows the evolution of the magnitudes for the years 2020-2023. All ss are quiet before the triggering event of 2020.07.22, except for s D (to some extent) (around day 7500 in the figure). To avoid overlaps in the plot, we displace the data of the sub-zones by adding a constant basis to the magnitude of each of them: +7 to sub-zone B, +14 to sub-zone C, and +22 to sub-zone D. Each sub-zone will be now

characterized separately.





Subzone A shows nearly no seisms previous to the $M_w7.8$ earthquake of 2020.07.22 and the magnitudes of the few previous events are less than 4.0. Then, near day 7500, the $M_w7.8$ seisms occur with no precursor, followed by a usual aftershock regime of seisms at short intervals and decreasing magnitude. However, sub-zones B or D do not reflect any of this activity. But just a few days after this triggering event (July 22, say), a swarm of quakes with magnitudes around and over 5.0 occurred in the

northeast sector of subzone D (Blue circles and orange triangles in Fig. 1). The upper part of Fig. 8 reflects this activity. It is not clear if the swarm in Sub-zone D is a consequence of the major earthquake in Sub-zone A, but it is a possibility.

Sub-zone B had weak previous activity, but the $M_w7.6$ earthquake of 2020.10.19 unleashed a continuous aftershock regime lasting until the last days of our recollection of data. Fig. 1 shows a "vertical" cluster of blue circles and orange triangles near latitude -159.5, showing this. This North-South activity spans all magnitudes from 2.1 to around 5.0. A recent example of the

latter is the 2023.10.09 $M_w5.0$ earthquake, just very close to the epicenter of earthquake B.

Sub-zone C had very weak activity before previous seisms; it did not show any response to neighboring seism A, but it follows the general aftershock regime generated by the 7.6 seism of 2020.10.19 in sub-zone B. Then, the sub-zone lost activity to almost none, just prior to the $M_w8.2$ earthquake that generated a modest aftershock activity here and in the neighboring A sub-zone to some extent. At present the activity of sub zone C is similar to that of sub zone A.

Subzone D had recovered 2021 its quiet behavior with almost no seismic activity until the $M_w7.2$ seism came in 2023, generating aftershock activity only here. These seisms are shown by blue circles and orange triangles to the East of the meridian at -160.5 in Fig. 1. Comparison to other sub-zones in Fig. 8 indicates only a possible connection to neighboring sub-zone B, near the corner 55N and -160.5W where a swarm of activity can be seen as blue circles and orange triangles in Fig. 8.

All of the previously described diversity in the seismic behavior is an indication of large differences in the subduction

processes. Even subzones only 100 km away (or even less) show completely different evolution, although they are wrapped under the common process of plate subduction. We will now examine this process from the optics of the entropy encoded in the sequence of data.

## 3.5 Mutability analysis

Fig. 9 shows an approximately constant initial mutability near value 0.62 beginning a descent near day 6 700, reaching a

minimum value at 0.60 to suddenly jump to higher levels near day 7 500. The filled red triangle indicates the position of the A earthquake in coincidence with the upward needle shown by the mutability function, which soon afterward decreases, evidencing the aftershock regime. However. Near day 7 900 a second needle in the mutability is appreciated: this is due to the $M_w8.2$ earthquake in the neighboring region C. From there, it goes through a true minimum from which it recovers soon remaining at low levels, which can mean a warning for future activity in this subzone.

Highly surprising is subzone B: despite having the largest amount of seisms among the 4 sub-zones, it has almost no activity prior to the $M_w7.6$ earthquake marked with a red solid triangle near day 7 600 in Fig. 10. The bare initial calculation yields a mutability of 0.61, from where the maximum corresponding to the seism B shows vigorously, to later fall because of the aftershock regime. The green hollow triangle marks the time of the C earthquake. The mutability function clearly marks the needle coinciding with earthquake B, and then it gradually decreases to 0.53 in the typical oscillatory pattern of aftershock

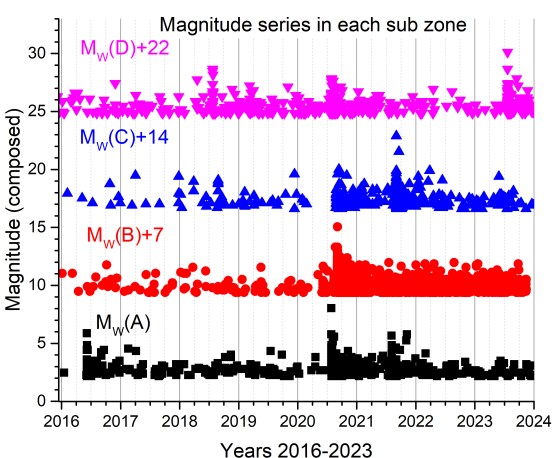

**Figure 8.** Magnitude of seisms of the sub-zones for the last 8 years. To avoid overwriting we added 7, 14, and 22 to the magnitudes of sub-zones B, C, and D respectively.

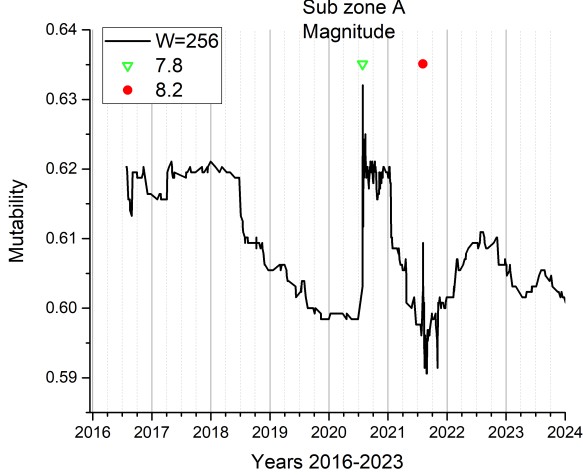

**Figure 9.** Mutability on magnitude for sub zone A.

regimes. Its recovery is far from being complete, remaining at levels near 0.58, which is even lower than present values for sub-zone A. The mutability function clearly shows the well-marked $M_w 8.2$, coinciding with earthquake B, and then it decreases to 0.53 in the typical oscillatory pattern of aftershock regimes.



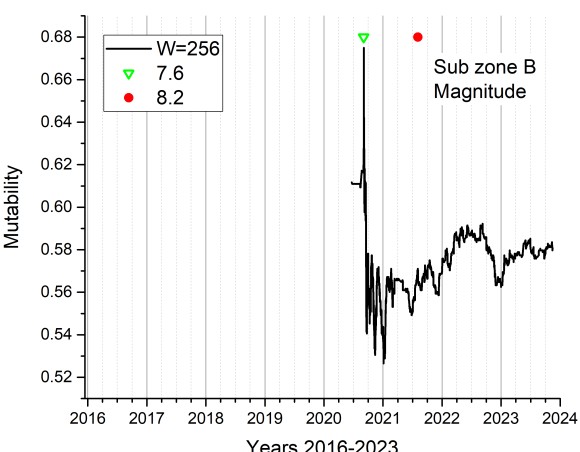

**Figure 10.** Mutability on magnitude for sub-zone B.

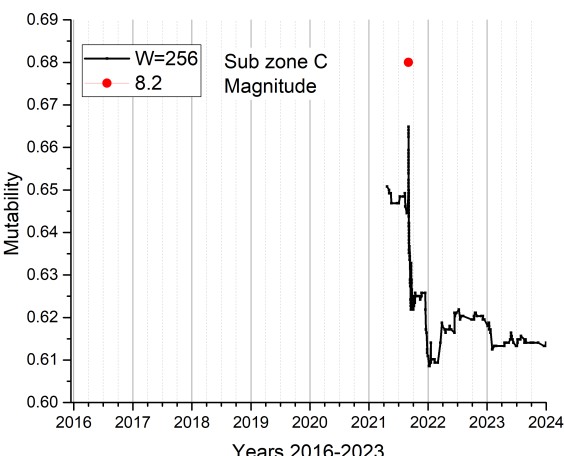

**Figure 11.** Mutability on magnitude for sub-zone C.

Subzone C has so few seisms that the curve begins just about 4 months before the largest $M_w 8.2$ earthquake. The needle of the mutability function coincides with the reed circle marking the time of seism C. The mutability function goes down remaining at values near 0.615, which is rather high.

Sub-zone D shows the expected needle at the position of the $M_w 7.2$ earthquake marked with a solid red triangle. The empty green triangle indicates the position of the $M_w 8.2$ earthquakes which is not shown in this plot. However, we find an unexpected broad maximum around day 7500. We had to go back to the time series to find out that this activity corresponds to a swarm


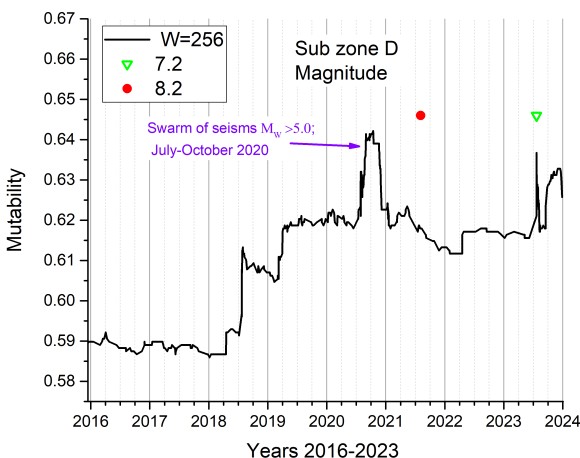

**Figure 12.** Mutability on magnitude for sub zone D.

of seisms with magnitudes between $M_w 5.5$ and $M_w 5.8$ happening in July 2018. The present level of the mutability function is

0.62, similar to sub-zones A, and C.

Despite the differences among sub zones, we can notice the sensitivity of mutability to detect the seismic activity in the time sequence. Although a premonitory protocol is still far away, we can recognize the decrease in the mutability value as one possible element to investigate further.

### 3.6    Time intervals

We can process the same previous data by using the series of time intervals (in minutes) between successive events. Fig. 13 presents the results for the 15 011 events over the 24 years, using two windows: one for $W =$256 consecutive events (black curve) and the other for $W =$512 consecutive events (red curve).

The series begins with intervals going from a couple of minutes to a thousand minutes or more; this means coverage of 3 to 4 orders of magnitude. Mutability looks for coincidences in registers, but there are very few chances of repeating previous values

and mutability values remain high (especially the one for the sorter window W). As the rocks underneath begin to break and accommodate, intervals shorten and coincidences increase, thus lowering the mutability function of intervals. At the critical points, shown by symbols in the upper right corner of this figure, we appreciate downward needles reaching extremely low values for mutability. This is a manifestation of the aftershock regime with successions of short intervals allowing a larger compression of the data in the window thus yielding low mutability values.

But Fig. 13 offers a surprise: the minimum reached by both curves at the beginning of the year 2016. It is clear that an important earthquake occurred in this region at this time. And sure enough, we find it: $M_w 6.2$, April 2, 2016. This is represented by the only red triangle on latitude 57N, on the upper right section of Fig. 1. A careful look at this portion of this plot reveals



circles of all diameters surrounding the red triangle, which correspond to the aftershock recovery, which extended for nearly a year, as shown by Fig. 13 in both window versions. Interestingly enough, this seism does not show up in a clear way

in the curves of Fig. 6, which is an indication of the higher sensitivity of the results obtained via intervals of consecutive seisms. However, we considered that working on magnitudes of the seisms is enough to present the main features of the seismic sequence and also because magnitude is the parameter at which the community looks the most. A complete analysis of intervals deserves a different framework, with data from different subduction areas to test generality and could be a line of future research.

Finally, Fig. 13 shows a general tendency to lower values of mutability on time intervals over these 24 years. The mutability has gone from nearly 1.2 ($W = 512$, red curve in Fig. 13) to about 0.8 at the end of 2023. This is a manifestation of a few repetitions in the data, intervals of all sizes, no clogging in the subduction process, and no large earthquake approaching from this point of view.

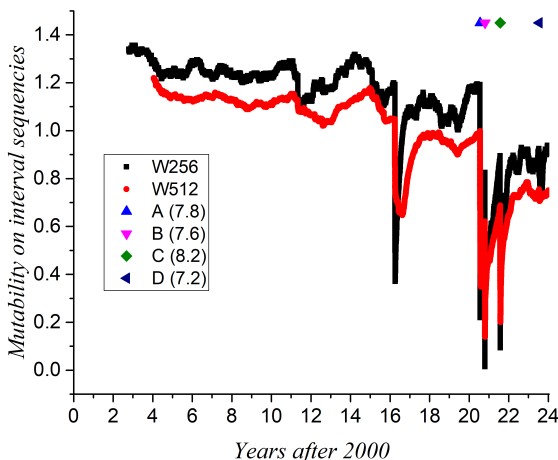

**Figure 13.** Mutability on the interval series along the 24 years of the present analysis.

## 4 Conclusions

We have analyzed the seismicity in the Alaska subduction zone measured during the last three years through non-extensive thermodynamics using two entropies: Tsallis entropy and mutability (or dynamical entropy). We have followed the time evolution of the seismic activity in the eastern and central zones of the Aleutian arc between the years 2000 and 2023, mainly focusing on four large earthquakes that occurred in that zone during the last years of the period under study.

We have used an entropic approach to deal with data produced by the seismic activity grouped in sequential series of $W$
consecutive events. Such non-overlapping time windows are analyzed to produce results on Tsallis entropy $S_q(t)$, which can



be summarized as follows. The dynamical values for $q(t)$ in the definition of the Tsallis entropy vary between 1.70 and 1.85, which is in accordance with values reported in the literature in other subduction areas (Telesca, 2010, 2011; Silva et al., 2006; Telesca and Chen, 2010; Darooneh and Mehri, 2010; Valverde-Esparza et al., 2012).

Tsallis entropy increases before seismic activity and collapses to minimal values at the instant of the megathrust. This is a manifestation of the pleiad of magnitude values brought in by the aftershock regime.

Mutability on magnitude values has precisely the opposite behavior as shown in Fig. 7. Thus, mutability minimizes before an incoming large earthquake since the magnitude of nearby seisms tends to be all small due to the clogging in the plate subduction. This accumulates stresses until a rupture occurs, producing chains of interrelated events (like a domino effect), which are described by non-additive entropies like the Tsallis entropy.

The definition of small sub-zones within the area of study leads to 4 sub-zones with similarities and differences. Thus, in all sub-zones, the frequency of inter events halted before a large ($M_w > 7.0$), but the level of activity is large (like in region D) or low (like in region B). So, even within a geographically restricted zone the subduction process is different for processes separated just for a few km.

Entropy (Tsallis or mutability) can be calculated for both magnitudes of seisms or intervals between consecutive seisms. A particular application of these techniques to a particular case can make use of the appropriate technique in due course.

The production of a large earthquake, like the $M_w$ 8.2 in sub-zone C, can occur in a quiet zone, without previous activity and without any previous indication. This requires permanent monitoring of the seismic activity of Alaska to find indicators of premonitory activity.

In the case of the subduction of the Nazca The behavior of this subduction zone is not comparable to the previous analysis of the subduction zone between the Nazca Plate and the South-American Plate, in this case, we did not find a behavior that warns about the coming of an earthquake, however, both the Tsallis entropy and the mutability show the state in which the study areas have been left, warning that the central area of the Aleutian arc is still active.

*Data availability.* https://earthquake.usgs.gov/earthquakes/search/

*Author contributions.* The authors are contributed equally to this manuscript.

*Competing interests.* The authors declare no competing interest.

*Acknowledgements.* Partial support from the following two Chilean sources is acknowledged: Fondecyt under contract 1230055, Financiamiento Basal para Centros Científicos y Tecnológicos de Excelencia (Chile) through the Center for Development of Nanoscience and Nanotechnology (CEDENNA) under contract AFB220001. This research has been partially supported by the Agencia Estatal de Investigación (grant no. PID2021-124701NBC21 y C22); the Universidad de Almería (grant no. FEDER/UAL Project UAL2020-RNM-B1980); the Consejería de Universidad, Investigación e Innovación, Junta de Andalucía (grant no. RNM104). PPITUAL, Junta de Andalucía-FEDER 2021-2027. Programa: 54.A. A.P., D.P. and E.E.V. have been partially funded by the Spanish Project LEARNIG PID2022-143083NB-I00 by the Agencia Estatal de Investigación.



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
