# Peer review of "2021 Alaska Earthquake: entropy approach to its precursors and aftershock regimes."

_Natural Hazards and Earth System Sciences, 2024_

## Referee Comment (RC3)

Referee's report on the manuscript nhess-2024-106

In this manuscript (ms) the authors use the Tsallis entropy and the mutability on seismic data from Alaska for understanding the seismicity in subduction zones. This work is interesting and within the scope of the journal. However, before its publication, the authors should consider the following improvements:

In Line 48, the authors write `` We include an Appendix …''. The appendix is missing from the ms.

Line 54, it is written `` … and −161.0∘…''. The limits for longitude should be written as for latitude.

Figure 2 should be improved to show the fitting of the data. The a and b values should also be given in the text.

In Line 72 Fig. 4 is mentioned. However, Fig. 3 is not yet discussed in the text. The authors should interchange the position of these figures.

In Line 84 `` mobile'' should be change to ``moving'' throughout the ms.

Say something about wlzip on line 111. Besides the authors refer to Pasten et al., 2023 for the technical details on the mutability, definitely a short description of this method in Subsubsection 2.2.1 will help the reader.

Studies for the seismicity of California and Japan by means of non-extensive statistical mechanisms have been published earlier by N.V. Sarlis, E.S. Skordas and P.A. Varotsos, Nonextensivity and natural time: The case of seismicity, *Phys. Rev. E* **82**, 021110 (2010). http://dx.doi.org/10.1103/PhysRevE.82.021110 .This paper is not mentioned in the first paragraph of page 8 lines138-147. The authors should also comment on the results of this paper for the sake of the reader's better information.

In the Section "Results and Discussion" in the Subsection 3.1 entitled "Tsallis Entropy", for the sake of the readers completeness of information the authors should discuss their results with respect to the very recent application of Tsallis entropy for improving the estimation of the occurrence time of an impending major earthquake (see the following two references):

P.A. Varotsos et al., Natural time analysis together with non-extensive statistical mechanics shorten the time window of the impending 2011 Tohoku M9 earthquake in Japan, Communications in Nonlinear Science and Numerical Simulation, 125 (2023) 107370. doi:10.1016/j.cnsns.2023.107370

P.A. Varotsos et al., Improving the Estimation of the Occurrence Time of an Impending Major Earthquake Using the Entropy Change of Seismicity in Natural Time Analysis. Geosciences, 13 (2023) 222. doi:10.3390/ geosciences13080222

In line 147 the authors should also refer to the very recent work of Flores-Márquez, E.L.;Ramírez-Rojas, A.; Sigalotti, L.D.G. Non-Extensive Statistical Analysis of Seismicity on the West Coastline of Mexico. Fractal Fract. 2024, 8, 306. https://doi.org/10.3390/fractalfract8060306 , along with some discussion since they also calculate q-values for the seismicity on various regions of the West Coastline of Mexico.

In view of the above, I will be glad to suggest publication of an appropriately revised manuscript.

---

## Author Comment (AC1)

**Answer to Referee 1**

I read careful the manuscript titled "2021 Alaska Earthquake: entropy approach to its precursors and aftershock regimes."

This work is very interesting for the scientific community because the study is focused to know more about the subduction zone in Alaska which has not been studied in the context of seismic activity.

However, in my opinion, the authors left important conceptual issues without consider deeply. In particular the physical interpretation about the mutability which is a concept not well known for everyone.

On the other hand, regarding with the format, there are many problems, in particular some figures aren´t set in the correct paragraphs and the captions do not explain correctly the important points in the graphs.

I think the manuscript requires a major revision in order to improve it.

I consider that the following issues must be taking into account by the authors.

Suggestions

1. **Section 2. In figure 2 the authors do no report the b-value and a-value estimated from the Gutenberg-Richter law. I suggest determine these values calculated yearly in order to search some pattern between the b-value and the largest magnitudes.**

   *Thank you for this comment. We have redrawn Fig. 2, including the linear adjustment yielding the overall values of parameters a and b.*

   *However, regarding yearly adjustments, we face the fact of the irregular distribution of seisms through the time scale. Thus, we go from a few dozen for some years to several thousand in more recent years. So, we believe it is difficult to compare any indicator yearly. Please see the Figure below. That is the reason we defined time windows of W events in natural time (succession of events), where you can get an idea of the time evolution of the process.*

[Figure]

2. **In line 72, authors cited first the Figure 4 but Figure 3 is not cited previously.**

   *We appreciate this comment. We apologize for this confusion. In the present version, we have rearranged these figures, and they are correctly cited now.*

3. **Relocate figures 3 and 4 close the paragraphs where they are cited.**

   *We have relocated the figures 3 and 4 close to their respective paragraphs.*

4. **In line 77, at the end, the authors cite an "Appendix" which does not appears in the manuscript.**

   *Yes, in a previous version there was an Appendix which we removed but one of the citations to it remained. Thank you for spotting it.*

5. **Figure 3 does not correspond with the description, in the text, lines 78-79.**

   *Thank you for this comment, we have carefully checked that all figures are with their descriptions.*

6. **In section 2.2 the authors describe the method used where they describe the Tsallis's entropy in Equation 4. However, to calculate the q-index the procedure is based on the fitting of Equation 10 (generalized Gutenberg-Richter) in the paper: "Flores-Márquez, E.L et al. Non-extensive statistical analysis of seismicity on the western coast of Mexico. Fractal fractal. 2024, 8, 306. https://doi.org/10.3390/fractalfract8060306." (This paper should be cited). The authors should explain the procedure they used to determine the q-index as well as present the fitting in Figure 2.**

   *Regarding this suggestion, referee #1 is correct and this important and recent Flores-Márquez, E.L et al. paper must be incorporated to clarify section 2. The wording we now propose is as follows (in red we highlight changes):*

   *"Eqs. 2, 3 and 4 are now treated analytically in the following way: states, and their probabilities p are formally treated as continuous variables, so integral replace sums. Lagrange multipliers combine equations 2, 3 and 4 to get conditions for maximum entropy. (The algebraic details have been recently given somewhere else (Posadas et al., 2023; Flores-Marquez et al., 2024) so we omit them here.) This leads to a q value determined by the distribution within each window which then leads to the Tsallis entropy by means of (Sotolongo-Costa and Posadas, 2004; Posadas and Sotolongo-Costa, 2023). In the present paper we make use of the second approach where Sarlis et al., (2010) or Telesca, (2010) approximation is used to get q parameter from b value. Once q is determined the Tsallis entropy calculation is straightforward:*

   *Equation [5]*

   *Boltzmann-Gibb's entropy is recovered in the limit $q \to 1$ but for most seismic zones $q > 1.0$ (interested readers are referred to Appedinx A in Flores-Marquez et al., 2024)."*

7. **Authors have to describe with more detail mutability concept (for example, as was described in: Eugenio E. Vogel et al., NHESS (2019) https://doi.org/10.5194/nhess-2019-309) as well as their physical interpretation.**

   *Thank you for this comment. We have added a more detailed description including a working example for the mutability in section 2.3.*

8. **In section 3.3. Caption of Figure 6 is not clear: it is written as "Left) Mutability on magnitude for the seismic sequence", but the y-label in Figure 6 Left appears "Sq" and, in Figure 6 Right, y-label indicate Mutability. Actually, the caption does not coincide with the Figure.**

   *Sorry for this mistake. Now the left panel is the right mutability curve intended.*

9. **Figure 8 is offside; it must be relocated within the section 3.4.**

   *Thank you for this comment, we have relocated the Figure 8.*

10. **The authors could design the Figure 8 as a panel with the four catalogues independently and indicating the EQ considered.**

    *Yes, we have followed your suggestion, and it is better now.*

11. **Results of Mutability reported in Figures 9 to 11 require a better explanation as well as their captions.**

    *We have added a more detailed descriptions on the captions of figures 9 to 11.*

12. **Finally, and according with the title, the authors must explain what about the precursors associated with the four earthquakes studied.**

    *We now point out the possible precursors along the discussions. In addition, we have included a special paragraph in the Conclusions to summarize this important point.*

    *Thank you for your time and positive comments.*

[revised manuscript text omitted]

---

## Author Comment (AC2)

**Answer to Referee 2**

Although the subject of the study is interesting, there are many points that need to be adequately addressed.

*OK.*

1) The English in the text is not fluent, resulting in misunderstandings in several sections. It requires thorough proofreading by a native English speaker.

*The language has been improved.*

2) Throughout the text, the authors appear to use the term "seismic risk" to refer to the "precursor" of large events. It is important to note that seismic risk is defined as the product of seismic hazard and vulnerability. Seismic hazard refers to the natural phenomena generated by an earthquake, while seismic risk is the probability of human loss or damage to the built environment when exposed to a seismic hazard.

*OK, we follow this precision and appropriate changes have been done through the text.*

3) In the Introduction, the authors state: "The aim of the present analysis is to find patterns in the data sequence that could lead to understanding the following aspects of the process: a) entropic activity of the zone or sub-zone that can be an indicator of seismic risk; b) parameters serving as indicators of seismic risk; c) behavior of the data sequence during the earthquake and immediate aftershocks; d) recovery to the 'normal' or previous seismic activity following the aftershock period." However, these patterns or analyses seem poorly discussed in the text.

*These points are now discussed in more details in Section 3. In particular, we now stress those aspects that could be considered precursors of important seismic activity. A paragraph dedicated to this aspect is now part of the conclusions.*

4) The description of data selection in Section 2.1 appears to be quite complicated. Initially, the authors extract earthquakes with a maximum depth of 100 km, then they re-select those with a maximum depth of 70 km. Why not select events with a maximum depth of 70 km from the beginning? Furthermore, there is another potentially more critical issue: the authors first apply the GR law to select all events (among those with a maximum depth of 100 km) with a magnitude larger than the completeness magnitude, then from this subset, they extract events with a maximum depth of 70 km. I believe the selection process should be different: first, they should select events with a maximum depth of 70 km, then apply the GR law to this dataset and consider only the events with a magnitude larger than the completeness magnitude.

*Yes, you are absolutely right. Several extractions were done during the preparation of this paper which caused the confusion. But now we go directly to the facts: what we use is an extraction with 19549 seisms which are used to get the GR plot (Fig. 2). From here we filter the data with magnitudes equal and over magnitude 2.1 and depth less or equal to 70 km, ending with a final catalog of 13871events.*

5)   The estimation of b-value is missing.

*Yes, in the present version the parameters of the adjustment to Gutenberg-Richter law, namely, a=5.70 and b=0.73, are explicitly given in the text and in Fig. 2.*

6)   Lines 72-73 "To the left, we appreciate how the activity of this region continuously increased irregularly since the beginning of the century": this is not "activity" (generally indicated by the number of events in a specified timescale), but the magnitude or the "energy".

*This sentence was changed according to your comment.*

7)   Lines 75-77: "At this point, we can follow two complementary ways to analyze these data through entropy: a) magnitude sequence, and b) sequence of intervals between consecutive seisms. We have chosen the former since entropy is more easily associated with energy, even since the early thermodynamics courses. However, we will include some results using the latter in the Appendix". Actually, this is quite a peculiar way to indicate their preference for analyzing magnitude instead of interevent times. Moreover, they mention that interevent times will be analyzed in the Appendix, which is missing!

*Yes, we decided now to leave out any discussion concerning the interevent times. We think this point needs to be studied under a broader scope, including different geographical areas, to compare it with the magnitude analysis. Some early results we have indicate that entropy on interevent catalogs can have advantages over the magnitude analysis, but this needs to be established properly. In a previous version, Fig. 13 was in an Appendix whose heading was removed but the reference to it remained. Sorry for that. The section on time intervals and previous figure 13 were removed from the present version.*

8)   Line 78: "In Fig. 3, the depiction presents the magnitude of each of the 15 011 seisms at the time of their production." Fig. 3 should be Fig. 4. And, please, rephrase: "Fig. 3 shows the time distribution of the analysed magnitudes".

*Yes, this was fixed. Figs. 3 and 4 were wrongly addressed in the previous version.*

9)   Lines 82-84: The sentence is really complicated. The authors can simply write "We calculated the Tsallis entropy and mutability in a moving window of size W, shifting by one event through the entire catalog and associate the calculated values to the time of the last event in the window".

*OK.*

10)   Line 90: "states" are simply "magnitudes".

*Done*

11) Line 94-98: why so complicated way to calculate the average magnitude? Just simply say that within each window you calculate the average magnitude.

*OK.*

12) I don't understand why the authors mention Shannon entropy (eq. 4) if it is never used in the study! While I understand the desire to shorten the theoretical treatment, the way the authors have done it makes the text very difficult to comprehend. It would be better to include all the details that allow the reader to understand how to calculate q.

*There are two different ideas in this comment. First, a referee for a previous paper insisted that we should mention Shannon entropy as another source of treating the data, so we included previous Eq. (4) to show the connection to our approach. Now we removed Eq. (4) and left just a comment (not a numbered equation) in the section devoted to Mutability. Second, since we had explicitly provided the calculation of $q$ in previous articles (Posadas and Sotolongo-Costa 2023 and Posadas et al. 2023.), we considered unnecessary to repeat the entire procedure here. However, we have included the most relevant part of the treatment in Section 2.2 guiding the interested reader to the appropriate literature.*

13) The same issue applies to the description of mutability. Mathematical details are required. Simply writing "the weight in bytes of the map created by wlzip of the time window at position $n$ n" leaves the reader unsure about what "weight," "wlzip," and "map" refer to.

*Since this concept is rather new, we have included details using a portion of the catalog defined here to illustrate the procedure and the definitions of the terms you mention. Although similar examples were included in previous papers, it is perhaps helpful to remind them with a close example. The subsection 2.3, "Mutability" contains all this material in the present version.*

14) Lines 119-123 are repeated in lines 131- 134.

*Yes, this was fixed.*

15) Line 152: "Fig. 5 reports the Tsallis' entropy calculated for windows W = 512 and W = 1024", Line 155: "Fig. 5 presents the average value of the Tsallis entropy over windows with W=512 and W=1024": is it the average value of Tsallis entropy or not??

*Sorry, there is nothing like instant values in our calculation. Results always correspond to average values depending on the initial time and the span of the time window. Thank you for noticing this misunderstanding.*

16) Lines 153-154: "The left-hand side presents the variations along the 24 years of this study, while the right-hand side zooms on the last 4 years, where most of the important activity shows up." This is repeated few lines below.

*Repetition was eliminated.*

17)    Line 158: "it can be noticed that at the precise moment the large earthquakes are produced, the Tsallis entropy suddenly decreases": better to write "The Tsallis entropy is characterized by a decrease before the occurrence of a large shock"

*Yes, we followed your suggestion.*

18)    Line 161: "to detect risk in advance" please see my comment in the point2)

*Yes, this was fixed.*

19)    Line 161: "poor statistics"??

*We rephrased this expression.*

20)    Line 162: "The average magnitude tends to be minimized before a large earthquake". Where is the figure showing this behavior of mean magnitude?

*This comment was eliminated.*

21)    Lines 165-175: the authors try to explain on a physical basis the behavior of Tsallis entropy, but honestly, I cannot follow their argument. They talk about 4 degrees latitude, 6 degrees longitude, stalagmites, stalactites, compressible gas…I would suggest to delete all this explanation, which, furthermore, lacks of adequate literature to be based on.

*OK, it was deleted.*

22)    In Line 158 "at the precise moment the large earthquakes are produced, the Tsallis entropy suddenly decreases."; in line 179 "Mutability decreases previous to a large earthquake because small seisms of restricted magnitude dominate the sequence", but in line 188 for W=256 "simultaneous increases of Tsallis entropy and decreases of mutability as the large earthquake approaches" the behavior of the two quantities is opposing. So, I should argue that the Tsallis and mutability have a correlated behavior for W=512 and 1024 and anticorrelated for W=256??? A DEEP explanation is NECESSARY.

*Yes, you are right. We are comparing two instances: previous to the earthquake (precursors) and at the moment of the earthquake and soon afterwards (aftershock). They became entangled. We have rewritten the entire paragraph.*

23)    Line 205: "We choose to present the time in number of days " but Figs. 8-12 shows years.

*Everything is in terms of years now.*

24)    Line 217: "Then, near day 7500," so, when?

*Now we refer to the middle of year 2020 instead.*

25)   Section 3.5 shows results only for W=256. At least for sub-zoness A and D, the results should also be presented for W=512 (due to what observed in my previous point 22).

*Yes, we followed your suggestion. It works well for subzone D, but not so well for subzone A. However, it serves to justify our discussion regarding to a minimum of records to do the statistics.*

26)   Fig. 13 should present also the results for W=1024.

*The whole discussion on inter-event times was deleted, so Fig. 13 is no longer in the paper.*

27)   A further figure is necessary showing the time variation of the Tsallis entropy Sq for the interevent times.

*Previous answer applies to this comment as well.*

*Let us say that we really appreciate your time and important comments, corrections and suggestions. Thank you so much.*

[revised manuscript text omitted]

---

## Author Comment (AC3)

**Answer to Referee 3**

In this manuscript (ms) the authors use the Tsallis entropy and the mutability on seismic data from Alaska for understanding the seismicity in subduction zones.

This work is interesting and within the scope of the journal. However, before its publication, the authors should consider the following improvements:

1.- In Line 48, the authors write `` We include an Appendix …''. The appendix is missing from the ms.

*Yes. There was a previous version with an Appendix; we removed it but a quotation to it remained. Thank you.*

2.- 54, it is written `` … and $-161.0\circ$…''. The limits for longitude should be written as for latitude.

*OK, it was done.*

3.- Figure 2 should be improved to show the fitting of the data. The a and b values should also be given in the text.

*OK. New version of Fig. 2 shows the regression with the fitting parameters.*

4.- In Line 72 Fig. 4 is mentioned. However, Fig. 3 is not yet discussed in the text. The authors should interchange the position of these figures.

*Sorry, there was a confusion in the presentation of Figs 3 and 4; this has been corrected in the new version.*

5.- In Line 84 `` mobile'' should be change to ``moving'' throughout the ms.

*Done.*

6.- Say something about wlzip on line 111. Besides the authors refer to Pasten et al., 2023 for the technical details on the mutability, definitely a short description of this method in Subsubsection 2.2.1 will help the reader.

*Well, the entire presentation of mutability was changed and enlarged providing technical details on mutability, including a working example.*

7.- Studies for the seismicity of California and Japan by means of non-extensive statistical mechanisms have been published earlier by N.V. Sarlis, E.S. Skordas and P.A. Varotsos, Nonextensivity and natural time: The case of seismicity, *Phys. Rev. E* **82**, 021110 (2010). http://dx.doi.org/10.1103/PhysRevE.82.021110 .This paper is not mentioned in the first paragraph of page 8 lines138-147. The authors should also comment on the results of this paper for the sake of the reader's better information.

*Thank you for this comment, we have added the reference and we have discussed their results in Section 3.1.*

8.- In the Section "Results and Discussion" in the Subsection 3.1 entitled "Tsallis Entropy", for the sake of the readers completeness of information the authors should discuss their results with respect to the very recent application of Tsallis entropy for

improving the estimation of the occurrence time of an impending major earthquake (see the following two references):

P.A. Varotsos et al., Natural time analysis together with non-extensive statistical mechanics shorten the time window of the impending 2011 Tohoku M9 earthquake in Japan, Communications in Nonlinear Science and Numerical Simulation, 125 (2023) 107370. doi:10.1016/j.cnsns.2023.107370

P.A. Varotsos et al., Improving the Estimation of the Occurrence Time of an Impending Major Earthquake Using the Entropy Change of Seismicity in Natural Time Analysis. Geosciences, 13 (2023) 222. doi:10.3390/ geosciences13080222

*Thank you for these two references, we have added both to the Section 2.2.*

9.- In line 147 the authors should also refer to the very recent work of Flores-Márquez, E.L.;Ramírez-Rojas, A.; Sigalotti, L.D.G. Non-Extensive Statistical Analysis of Seismicity on the West Coastline of Mexico. Fractal Fract. 2024, 8, 306. https://doi.org/10.3390/fractalfract8060306 , along with some discussion since they also calculate q-values for the seismicity on various regions of the West Coastline of Mexico.

*Thank you for this suggestion. We have added the reference and we have discussed it in Section 2.2.*

In view of the above, I will be glad to suggest publication of an appropriately revised manuscript.

*Thank you for your time and appropriate corrections and suggestions.*

[revised manuscript text omitted]